# Tissue Characterization Using an Electrical Bioimpedance Spectroscopy-Based Multi-Electrode Probe to Screen for Cervical Intraepithelial Neoplasia

**DOI:** 10.3390/diagnostics11122354

**Published:** 2021-12-14

**Authors:** Tong In Oh, Min Ji Kang, You Jeong Jeong, Tingting Zhang, Seung Geun Yeo, Dong Choon Park

**Affiliations:** 1Department of Biomedical Engineering, College of Medicine, Kyung Hee University, Seoul 02447, Korea; tioh@khu.ac.kr (T.I.O.); yj.jeong0107@gmail.com (Y.J.J.); zttouc@hotmail.com (T.Z.); 2Medical Science Research Institute, Kyung Hee University Medical Center, Seoul 02447, Korea; yeo2park@gmail.com; 3Department of Obstetrics and Gynecology, College of Medicine, The Catholic University of Korea, Seoul 06591, Korea; star6777@naver.com; 4Department of Obstetrics and Gynecology, Saint Vincent’s Hospital, The Catholic University of Korea, Suwon 16247, Korea

**Keywords:** CIN, electrical impedance spectrum, cancer screening, bioimpedance spectroscopy, multi-electrode screening probe

## Abstract

The successful management of cervical intraepithelial neoplasia (CIN) with proper screening and treatment methods could prevent cervical cancer progression. We propose a bioimpedance spectroscopic measurement device and a multi-electrode probe as an independent screening tool for CIN. To evaluate the performance of this screening method, we enrolled 123 patients, including 69 patients with suspected CIN and 54 control patients without cervical dysplasia who underwent a hysterectomy for benign disease (non-CIN). Following conization, the electrical properties of the excised cervical tissue were characterized using an electrical bioimpedance spectroscopy-based multi-electrode probe. Twenty-eight multifrequency voltages were collected through the two concentric array electrodes via a sensitivity-optimized measurement protocol based on an electrical energy concentration method. The electrical properties of the CIN and non-CIN groups were compared with the results of the pathology reports. Reconstructed resistivity tended to decrease in the CIN and non-CIN groups as frequency increased. Reconstructed resistivity from 625 Hz to 50 kHz differed significantly between the CIN and non-CIN groups (*p* < 0.001). Using 100 kHz as the reference, the difference between the CIN and non-CIN groups was significant. Based on the difference in reconstructed resistivity between 100 kHz and the other frequencies, this method had a sensitivity of 94.3%, a specificity of 84%, and an accuracy of 90% in CIN screening. The feasibility of noninvasive CIN screening was confirmed through the difference in the frequency spectra evaluated in the excised tissue using the electrical bioimpedance spectroscopy-based multi-electrode screening probe.

## 1. Introduction 

Cervical intraepithelial neoplasia (CIN), also called cervical dysplasia, is a precancerous lesion of cervical cancer in which epithelial cells of the cervix become dysplastic. These local lesions, limited to the cervical epithelium, can progress to cervical cancer due to various factors, including human papillomavirus (HPV) infection [1]. CIN can be evaluated histologically and divided into three grades (I, II, and III) based on the thickness of the epithelium containing the dysplastic cells. Low-grade squamous intraepithelial lesions (LSILs, also known as CIN I) are now recognized as a histological diagnosis of benign viral replication that should be managed conservatively, whereas CIN III is recognized as a true pre-invasive precursor with the potential to progress to cancer [2]. It has been estimated that, in 70% of affected women, CIN II–III can persist or progress to cervical cancer after 10–20 years [2,3]. Therefore, diagnosis and treatment at the stage of CIN may prevent the development of cervical cancer. 

At present, cervical dysplasia in women suspected of having lesions from a Papanicolaou (Pap) smear or colposcopy examination is diagnosed by biopsy. However, the Pap smear test has relatively low sensitivity, although, it has the advantages of being inexpensive and simple [4]. Diagnosis by colposcopy and punch biopsy requires a gynecological expert. Alternatively, conical resection can be used for both diagnosis and treatment, but this method is highly invasive. Moreover, this method can be deleterious in young women, especially those who want to become pregnant, as it can increase the likelihood of premature birth due to cervical incompetence. Despite these early screening tools, however, the high false-negative rate of existing diagnostic methods has limited the early diagnosis of cervical cancer [1]. In addition, current diagnostic procedures, such as punch biopsy and LEEP conization, are invasive, and analysis of their results takes a long time. For these reasons, it is clinically required to develop a noninvasive, accurate, and inexpensive method for diagnosing CIN. It would be beneficial to minimize the waiting time from screening to diagnosis and even treatment, if we can obtain the result immediately. Moreover, it could reduce the number of patients who must undergo conical resection for diagnosis. 

The impedance spectra of living tissues vary due to the differences in morphological structures and components. Therefore, the resistivity spectrum measured at each frequency is highly related to the physiological and pathological characteristics of the tissue [5,6,7]. Bioimpedance spectroscopy, a method of measuring electrical properties over a wide range of frequencies, can characterize target tissue. The advantages of bioimpedance spectroscopy include its low cost, high sensitivity, ability to diagnose patients in real-time, and it is radiation-free [8]. Brown et al. reported that the electrical impedance spectra on the cervix could separate normal and CIN [9]. Thus, bioimpedance spectroscopy can potentially facilitate cancer screening [10], thereby reducing the incidence of cervical cancer among women in developing countries. Subsequently, many studies have evaluated whether CIN can be diagnosed based on the electrical properties of cervical tissues [9,10,11,12]. In addition, clinical studies using four electrodes with colposcopy have been performed to measure the bioimpedance spectra of suspected local regions of the cervix [13,14,15,16]. However, these methods were performed as auxiliary tools to improve the sensitivity and specificity in the suspected focal region as CIN, in addition to conventional cervical cancer screening tests. 

In this study, we propose electrical bioimpedance spectroscopy with a multi-electrode probe as an independent noninvasive CIN screening method for covering the region, including the squamous-columnar junction of the cervical tissue. To evaluate the performance of the CIN screening method, we used a bioimpedance spectroscopic measurement device and a multi-electrode screening probe to measure the electrical properties of excised cervical tissue samples. The associations between these electrical properties and the histological characteristics of the cervix were analyzed. We focused on scientific and technical proof to detect changes in bioimpedance due to histological changes and distinguish diseases rather than classification according to the clinical significance of each CIN I, II, and III. Moreover, we evaluated whether these differences in electrical properties can be exploited to noninvasively screen for CIN. 

## 2. Methods

### 2.1. Subjects

This study was approved by the institutional review board of St. Vincent’s Hospital of The Catholic University of Korea, and informed consent was obtained from each patient (VC17TNSI0126, VC20TISI0072). The 123 women, aged from their 20s to 60s, comprising 69 with suspected CIN who underwent loop electrical excision procedure (LEEP) and 54 without CIN who underwent hysterectomy for benign disease, participated in this study. In this study, the group of CIN I was composed of patients who were suspected of CIN and diagnosed with CIN I from LEEP. Patients with CIN I who underwent LEEP were as follows; patients with ASC-H or high-grade intraepithelial lesion (HSIL) in colposcopy but CIN I on cervical biopsy, patients with suspicious CIN II/III in colposcopy but their final diagnosis was CIN I by LEEP, and patients who refused ECC (endocervical curettage) or had cancer phobia.

The sizes of all cervical tissues were recorded, and all samples were photographed before measurement. The cervical tissue was gently cleaned with normal saline gauze to remove mucous discharge and/or blood. The electrical properties of the tissue were measured on a plate that did not conduct electricity. The probe holder was used to ensure stable contact between the probe and tissue during measurements. The probe was initially positioned to contact the entire surface of the cervical tissue and then fixed to the probe holder to maintain the position of the probe.

### 2.2. Structure of the Multi-Electrode Screening Probe

The bioimpedance of the cervical tissue was measured using a bioimpedance spectrum-based screening device and a multi-electrode screening probe (Figure 1). The probe was 10 mm in diameter, which determined a suitable size to include the squamous-columnar junction of the cervical tissue. Seventeen small, gold-plated circular electrodes were attached to the probe, with 16 of these electrodes arranged in two circular rings for current injection and voltage measurement and one ground electrode placed at the center of the probe. Each of the 17 electrodes was attached to a small spring-loaded pin, 0.9 mm in diameter, which minimized errors by improving contact between the electrodes and tissue due to curvature of the surface of tissue. In addition, the structure containing each small spring-loaded pin was designed such that contaminated parts could be easily replaced. The mechanical part of the multi-electrode screening probe, including the signal processing board, was constructed and assembled (Figure 2a). Because the assembly nut and header part constituted the contact between the electrode and tissues, they were composed of biocompatible materials, as evaluated by the ISO 10993-1:2009 [17]. Each electrode was wrapped in an electrode sealing rubber pad to reduce the effects of liquids and external substances on the electrode surfaces.

### 2.3. Bioimpedance Spectrum-Based Screening Device 

The bioimpedance spectroscopic measurement device for CIN screening was developed by using 16 measurement electrodes and operating frequencies from the KHU Mark2.5 EIT (electrical impedance tomography) system [18]. It consisted of two impedance measurement modules (IMM) for the configuration of four-electrode impedance measurement among multi-electrode pins. Each IMM consisted of a constant current source and a voltmeter. The bioimpedance measurement system generated a balanced current source from the two IMMs, received a control signal from a controller board to calculate impedance, injected a current into a subject, and measured the induced voltage.

The Field-Programmable Gate Array (FPGA; EP3C10F256C8N, Altera, San Jose, CA, USA) inside the IMM was responsible for regulating an impedance measurement control device, a digital waveform generator, a digital phase-sensitive demodulator, a communication module, a timing control module, a gain control module, and a calibration module. It worked with a digital–analog converter that receives data from a digital waveform generator on the FPGA and generates an analog voltage waveform to output a constant current source. The IMM included a voltage measurement circuit and an analog-to-digital converter for measuring a differential mode voltage signal. To measure the electrical properties of the excised cervical tissue, the analog voltage waveform generated by the digital–analog converter was converted to the current through a voltage–current conversion circuit inside the IMM and injected into the targeted tissue through an arbitrary pair of electrodes according to the measurement protocol. Next, the voltage signal induced by the injected current was measured through a differential signal amplifier, passed through an analog amplifier stage, adjusted to a signal size suitable for the input range of the analog–digital converter, and converted to a digitized signal. After preprocessing, the voltage data inside the FPGA were transferred to the microcontroller in the controller board according to the internal operation sequence.

Among 16 electrodes, one pair of electrodes was responsible for the current injection, and the other pair measured the induced voltage. The current was injected without saturation at six frequencies, 0.625, 1, 5, 10, 50, and 100 kHz, with a peak–to–peak magnitude ranging from 85 to 305 µA. This measurement frequency range included information distinguishing normal from CIN in studies based on the cervical tissue model [19]. Inside the multi-electrode screening probe, a signal processing circuit was implemented to amplify the measurement signal and increase the input impedance of the measurement electrodes by using high precision and low noise rail–to–rail amplifiers (OPA4140, Texas Instruments, Dallas, TX, USA). The signal processing board of the probe was designed in a multilayered board structure to minimize the probe size (Figure 2b).

Induced voltage was multiplied by the reconstructed matrix derived from the sensitivity analysis of the electrical impedance tomography (EIT) to reconstruct the resistivity at each measurement frequency [20]. The reconstructed resistivity spectra were representative of the electrical properties of the tissue samples underneath the probe. The classification of cervical tissue samples based on their measured spectra was compared with their pathological diagnosis.

### 2.4. Calibration of and Measurements with the Screening Probe

Before each tissue measurement, the screening probe was cleaned thoroughly with saline and alcohol to remove blood and mucus. The screening probe was calibrated by immersion in saline of known electrical conductivity. Measured voltage data were used to calibrate the reconstructed resistivity and confirm the stability of the system. The bioimpedance of cervical tissue samples was measured in the operating room immediately after removal by LEEP conization or hysterectomy. The probe was positioned so all 17 spring-loaded pin electrodes were in contact with the tissue sample. The measurement quality was assessed by comparing the impedance trend resulting from the difference in position of the electrode pairs. Each tissue sample was measured three times to check reproducibility, with less than 1 minute required for each set of measurements. The results of the three assessments were averaged, and the resistivity spectrum results at each frequency were determined. Figure 3 shows the experimental setup for measuring a cervical tissue sample.

### 2.5. Statistical Analysis

The statistical analysis was performed between the non-CIN group and CIN group, and the independent *t*-test was used for the comparison of demographic and clinical characteristics. The magnitude of reconstructed resistivity spectra between groups was statically analyzed, and the results were expressed as mean ± SD. The association between frequency spectra and the pathological state of tissue was compared by a paired *t*-test or a two-sample independent *t*-test. The *p*-value < 0.05 was considered to indicate a statistically significant difference. 

## 3. Results

The 123 enrolled patients consisted of 25, 15, and 29 patients diagnosed with CIN grades I, II, and III, respectively, and 54 patients without CIN (non-CIN group). The baseline demographic and clinical characteristics of these patients are shown in Table 1.

The electrical properties of the cervical tissue samples were measured precisely using a bioimpedance spectrum-based screening device with a multi-electrode screening probe. Each tissue sample was assessed three times. The average standard deviation from all frequencies was 0.057 ± 0.042 Ω. Overall, the reconstructed resistivity tended to decrease as the frequency increased, with comparisons between frequencies being significantly greater for samples from patients without than with CIN (Figure 4a,b). The difference in electrical properties between the non-CIN and CIN groups was pronounced, differing significantly at reconstructed resistivities ranging from 625 Hz to 50 kHz (*p* < 0.05), but not at a reconstructed resistivity of 100 kHz (*p* = 0.9787; Table 2). The reconstructed resistivity at each frequency was analyzed using 100 kHz as the reference (Figure 4c,d). The differences between the non-CIN and CIN groups were analyzed using paired *t*-tests’ whereas, the reconstructed resistivity at each frequency was analyzed as an independent variable using two-sample independent *t*-tests. In both analyses, the differences between the non-CIN and CIN groups were statistically significant (*p* < 0.001; Table 3). The HPV (*p* < 0.0002) and conization size (*p* < 0.0001) also significantly differed between the two groups. However, there were no significant differences in age (*p* = 0.36), gravida (*p* = 0.68), BMI (*p* = 0.3), menopause (*p* = 0.87), and cervicitis (*p* = 0.37) between the two groups. 

Using logistic regression analysis, differences in reconstructed resistivity at each frequency relative to that at 100 kHz were used to calculate the sensitivity and specificity based on the area under the difference curve.

To validate these findings, cutoff values were calculated using the receiver operating characteristic (ROC) curve. Patients with values above and below the estimated cutoff value were compared by *t*-test. An area under the ROC curve of 0.90 resulted in a sensitivity of 94.3% and a specificity of 84%. Significant results were obtained at all frequencies, with the optimal results obtained when comparing frequencies of 625 Hz and 100 kHz.

## 4. Discussion

The successful management of CIN depends on proper screening and accurate diagnosis. The determination of electrical properties may be diagnostic of cervical dysplasia and may overcome the limitations of visual diagnosis. In particular, lesions that are difficult to observe with the naked eye or by colposcopy, such as isolated or endocervical lesions, may be diagnosed by non-visual methods, thereby increasing the accuracy of diagnosis. In addition, the diagnosis of cervical dysplasia using electrical properties is objective, enabling diagnosis by a non-gynecologist and reducing the frequency of referral for cervicography or colposcopy. 

Since biological tissues contain components with both resistive and capacitive properties, the impedance magnitude on the frequency is related to the organization of the tissue. The cervical epithelium is a highly structured tissue of stratified cells [21,22]. The development of cervical dysplasia from normal tissue is accompanied by structural changes caused by an increased nuclear/cytoplasmic ratio, loss of the layer of flattened cells close to the surface, and surface mucus and stromal tissue characteristics [9,23]. The destruction of tissue structure could reduce resistivity at low frequencies, making the difference between normal and CIN samples particularly apparent at low frequencies [9]. The present study results confirmed that the resistivity differences between non-CIN and CIN were more apparent in the frequency spectrum. 

Additionally, the resistivity differences between non-CIN and CIN groups showed consistent results regardless of the patient’s clinical features, namely, age, gravida, BMI, menopause, and cervicitis. However, there was a significant association between HPV infections or conization size and the CIN state, as observed in this study. In the case of conization size, this difference could be induced because the size of the excised sample was also quite different due to LEEP or hysterectomy, as shown in Table 1. Despite the difference in sample size, the high sensitivity and specificity resulting in the screening process using the resistivity spectrum raise expectations for future clinical application. However, we should pay more attention to subjects with HPV infections. It was unclear whether HPV directly influenced the resistivity spectrum or the deformed tissue affected the results since CIN was mostly accompanied by HPV infections.

In previous studies, the measurements of the impedance spectra of cervical tissue using a four-electrode probe showed the feasibility of assessing the progress of cervical dysplasia [10,24]. However, the sensitive region of the four-electrode measurement method was limited to a local region due to the size of the electrodes. When extending the distance among four electrodes to measure the electrical property in a larger area, the sensitivity and accuracy were reduced. To overcome these limitations, the changes in the electrical properties of the cervix were measured in the present study using an impedance spectrum-based multi-electrode probe. The probe, designed to detect changes in the electrical properties of the transformation zone of the cervix, consisted of one central ground electrode and 16 radially distributed gold-plated electrodes for current injection and voltage measurement. This overcame the limitations of the four-electrode probes, including those associated with the position of the electrode, the structure of the surrounding tissue, and the inability to screen a large area [25]. Furthermore, the diameter of the multi-electrode probe was set at 1 cm, enabling a diagnosis of the postmenopausal atrophic cervix or nulliparous women. This method has many advantages as a potential screening test. Unlike conventional diagnostic devices, a multi-electrode screening probe is noninvasive and can rapidly and easily measure a broad area of the cervix simultaneously. Such a screening tool will be helpful in many countries in which pathology reports cannot be provided rapidly.

When normal tissue progresses to cervical dysplasia, the stromal contribution to tissue resistivity is influenced by the nuclear/cytoplasmic ratio, surface mucus and stromal tissue characteristics, and cellular density [9]. This study found that the measured impedance spectra differed in cervical dysplasia and normal tissue. In addition, probe performance and its applicability as a cervical cancer screening tool were confirmed, as shown by the agreements between impedance spectra and pathologic results. The use of this multi-electrode CIN screening probe resulted in the differentiation of non-CIN from the dysplastic squamous epithelium and of CIN I from CIN II/III tissues. A comparison of several methods for diagnosing squamous intraepithelial lesions reported that the areas under the ROC curves were 0.76 for Pap smear testing and 0.84 for diagnostic colposcopy [26]. Our method showed that an area under the ROC curve of 0.90 resulted in a sensitivity of 94.3% and specificity of 84%. The advantage of this method as a potential screening test is its ability to provide an immediate result, enabling further investigation or treatment in patients with high-grade CIN.

Several new technologies have been proposed to detect cervical cancer [27,28,29,30]. When comparing our proposed method to them, there were limitations of our study. Firstly, although one of the advantages of the multi-channel electrode probe is its ability to measure a wide area simultaneously, it measures the mean of the electrical properties of the tissue. It means that CIN lesions may be difficult to detect when their sizes are much smaller than the measured area. Therefore, even after being diagnosed with CIN, the treatment area may be larger than necessary. Considering these issues, we could suggest applying electrical property imaging. Recently developed EIT techniques have enabled noninvasive and accurate measurement of the electrical properties of biological tissues. These methods may be simpler and less expensive than conventional medical imaging devices, such as MRI, CT, and ultrasound. In addition, our finding that differences in electrical properties correlated with the degree of dysplasia of cervical epidermal tissue can allow electrical property images to provide information about lesion degree and location. Secondly, the electrical properties of cervical tissue were obtained after LEEP or hysterectomy. The results could be varied even though we immediately measured after resecting the tissue in the operating room. To measure the electrical properties of cervical tissue in situ before resection would require a change in the design of the instrument, such as changes in the diameters and lengths of the probe or handle. These modifications of the probe and system for the measurement of domain-specific electrical properties will likely enhance the clinical use of this method and enable minimally invasive treatment. Finally, the performance of the probe will be further improved and must be validated through in vivo clinical studies. We intend to conduct an additional analysis that helps clinical treatment decisions by distinguishing between HSIL and LSIL. The relation between bioimpedance spectrum and clinical significance, such as the progression of the lesion and the prognosis, could provide more clinical evidence for determining treatment methods.

## 5. Conclusions

A bioimpedance spectroscopic measurement device and a multi-electrode probe were used to screen for CIN on the excised cervical tissue in this study. The multi-electrode probe was large enough to measure the electrical property of the squamous–columnar junction of the cervical tissue to use it as an independent screening tool for CIN. Reconstructed resistivity showed a significant difference between the CIN and non-CIN groups when 100 kHz was used as the reference frequency. Determination of cervical electrical properties provides a potentially promising screening tool with a sensitivity of 94.3% and specificity of 84%, similar to those of currently used screening tests. Moreover, applying the bioimpedance spectrum-based multi-electrode probe can provide instant resistivity spectrum results for screening for CIN. It is necessary to develop a classification algorithm that can improve the accuracy of the screened results by using the measured complex impedance data. Moreover, it must be verified through in vivo clinical studies.

## Figures and Tables

**Figure 1 diagnostics-11-02354-f001:**
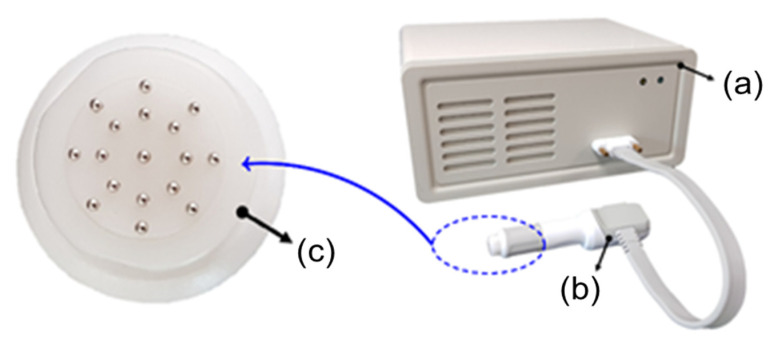
The system for screening of CIN using electrical bioimpedance spectroscopy-based multi-electrode probe: (**a**) bioimpedance spectroscopic measurement device, (**b**) multi-electrode probe, and (**c**) configuration of multiple pin electrodes used for the screening probe.

**Figure 2 diagnostics-11-02354-f002:**
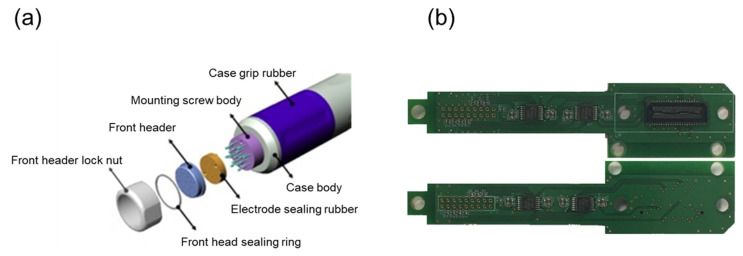
Illustration of the multi-electrode screening probe: (**a**) Case design and (**b**) Multi-layered signal processing board.

**Figure 3 diagnostics-11-02354-f003:**
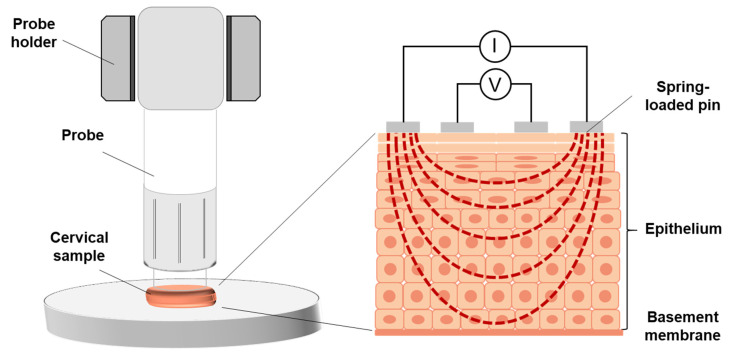
Schematic representation of the experimental setup.

**Figure 4 diagnostics-11-02354-f004:**
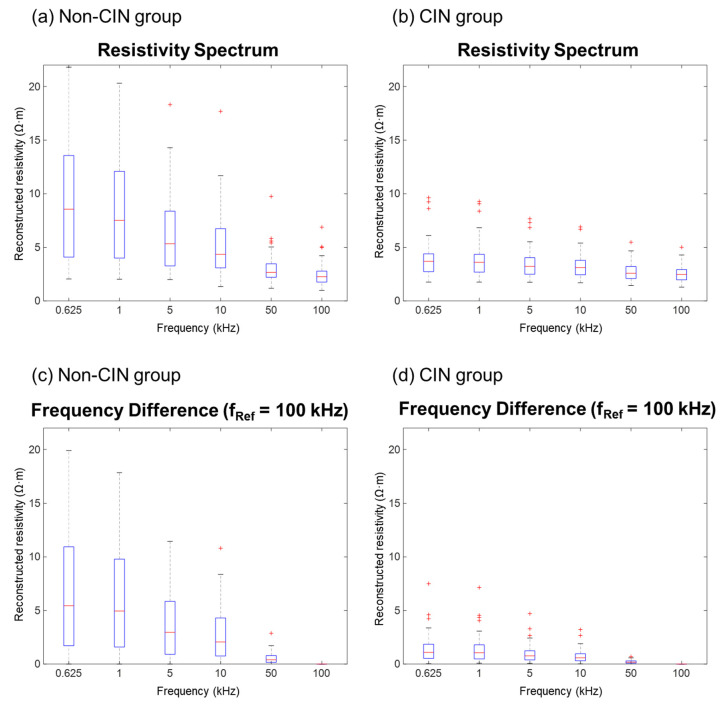
Resistivity and frequency difference resistivity spectra. (**a**,**b**) Resistivity as a function of frequency in cervical tissues from the (**a**) non-CIN and (**b**) CIN groups. (**c**,**d**) Difference in resistivity at each measured frequency relative to that at 100 kHz in cervical tissues from the (**c**) non-CIN and (**d**) CIN groups.

**Table 1 diagnostics-11-02354-t001:** Demographic and clinical characteristics of the patients included in this study.

	Non-CIN (n = 54)	CIN I (n = 25)	CIN II (n = 15)	CIN III (n = 29)
Age (median (min–max)), yr	48 (28–78)	47 (27–78)	47 (21–74)	42 (18–71)
Gravida				
Multigravida	44	20	12	23
Nulligravida	10	5	3	6
BMI (mean ± SD), kg/m^2^	24.35 ± 3.81	23.07 ± 3.46	23.44 ± 8.47	24.72 ± 3.87
Menopause	18	12	5	11
Positive for high risk HPV	31	16	13	15
Conization size (diameter)				
≤2 cm	10	18	3	19
>2 cm to ≤3 cm	19	4	11	7
>3 cm	25	3	1	3
Cervicitis	18	25	15	29

Abbreviations: CIN, cervical intraepithelial neoplasia; BMI, body mass index; HPV, human papillomavirus.

**Table 2 diagnostics-11-02354-t002:** Reconstructed resistivities of cervical tissues from the non-CIN and CIN groups.

Frequency [kHz]	Reconstructed Resistivity [Ω·m]		
	Non-CIN (n = 54)	CIN (n = 69)	*p*
0.625	10.73 ± 5.35	3.92 ± 1.62	<0.0001
1	9.95 ± 4.95	3.85 ± 1.60	<0.0001
5	6.98 ± 3.42	3.46 ± 1.28	<0.0001
10	5.76 ± 3.00	3.26 ± 1.14	<0.0001
50	3.14 ± 1.46	2.72 ± 0.86	0.0499
100	2.51 ± 1.08	2.50 ± 0.77	0.9787

Abbreviation: CIN, cervical intraepithelial neoplasia.

**Table 3 diagnostics-11-02354-t003:** Difference in reconstructed resistivity relative to 100 kHz of cervical tissues from the non-CIN and CIN groups.

	Paired *T*-Test		Two-Sample Independent *T*-Test		
	Difference ± SE [Ω·m]	*p*	Difference ± SE [Ω·m]		*p*
			Non-CIN (n = 54)	CIN (n = 69)	
625 Hz	4.25 ± 0.45	<0.0001	8.22 ± 5.33	1.42 ± 1.28	<0.0001
1 kHz	3.89 ± 0.41	<0.0001	7.44 ± 4.87	1.35 ± 1.25	<0.0001
5 kHz	2.42 ± 0.24	<0.0001	4.47 ± 3.03	0.96 ± 0.83	<0.0001
10 kHz	1.80 ± 0.19	<0.0001	3.26 ± 2.44	0.76 ± 0.62	<0.0001
50 kHz	0.39 ± 0.04	<0.0001	0.63 ± 0.53	0.22 ± 0.16	<0.0001
Area **			8.38 ± 5.28	1.72 ± 1.46	<0.0001

** The area under the receiver operating characteristic (ROC) curve was calculated based on the difference in reconstructed resistivity relative to that at 100 kHz.

## Data Availability

The data presented in this study are available on request from the corresponding author.

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
