# Peer review of "Tissue Characterization Using an Electrical Bioimpedance Spectroscopy-Based Multi-Electrode Probe to Screen for Cervical Intraepithelial Neoplasia"

_diagnostics, 2021, doi:10.3390/diagnostics11122354_

Round 1
Reviewer 1 Report
Please see the attachment.

Author Response
Please see the attachment response and revised paper together.

Reviewer 2 Report
The study by Tong In Oh et al. aimed to evaluate the ability of electrical bioimpedance spectroscopy to non-invasively diagnose cervical intraepithelial neoplasia. Overall, the study is well organized and well written however some concerns need to be raised.
- The information on Ethical approval should be put into the Methods section.
- It is not possible to assess statistical analysis you performed. Please add subsection on description of statistics used into the Methods section.
- Table 1: together with median show min-max, and together with mean value of BMI show standard deviation.
- Table 1: you should statistically compare all subgroups in terms of all parameters.
- In all tables leave "p" instead of "p value".
- The limitations of the research should be presented at the end of Discussion.
Author Response
The study by Tong In Oh et al. aimed to evaluate the ability of electrical bioimpedance spectroscopy
to non-invasively diagnose cervical intraepithelial neoplasia. Overall, the study is well organized and
well written however some concerns need to be raised.
1. The information on Ethical approval should be put into the Methods section.
Answer: We included ethical approval information in section 2.1 on page 2.
Changed Manuscript:
On page 2, “This study was approved by the institutional review board of St. Vincent’s Hospital of The
Catholic University of Korea, and informed consent was obtained from each patient (VC17TNSI0126,
VC20TISI0072).”
2. It is not possible to assess statistical analysis you performed. Please add subsection on description
of statistics used into the Methods section.
Answer: We added to section 2.5 on page 5, which presented statistical analysis.
Changed Manuscript:
On page 5, “The statistical analysis was performed between the non-CIN group and CIN group, and the
independent t-test was used for the comparison of demographic and clinical characteristics. The
magnitude of reconstructed resistivity spectra between groups was statically analyzed, and the results
were expressed as mean ± SD. The association between frequency spectra and the pathological state
of tissue was compared by a paired t-test or a two-sample independent t-test. The p-value<0.05 was
considered to indicate a statistically significant difference.”
3. Table 1: together with median show min-max, and together with mean value of BMI show standard
deviation.
Answer: We improved table 1 on page 6 as recommended.
Comments:
Reviewer 2:
The study by Tong In Oh et al. aimed to evaluate the ability of electrical bioimpedance spectroscopy
to non-invasively diagnose cervical intraepithelial neoplasia. Overall, the study is well organized and
well written however some concerns need to be raised.
1. The information on Ethical approval should be put into the Methods section.
Answer: We included ethical approval information in section 2.1 on page 2.
Changed Manuscript:
On page 2, “This study was approved by the institutional review board of St. Vincent’s Hospital of The
Catholic University of Korea, and informed consent was obtained from each patient (VC17TNSI0126,
VC20TISI0072).”
2. It is not possible to assess statistical analysis you performed. Please add subsection on description
of statistics used into the Methods section.
Answer: We added to section 2.5 on page 5, which presented statistical analysis.
Changed Manuscript:
On page 5, “The statistical analysis was performed between the non-CIN group and CIN group, and the
independent t-test was used for the comparison of demographic and clinical characteristics. The
magnitude of reconstructed resistivity spectra between groups was statically analyzed, and the results
were expressed as mean ± SD. The association between frequency spectra and the pathological state
of tissue was compared by a paired t-test or a two-sample independent t-test. The p-value<0.05 was
considered to indicate a statistically significant difference.”
3. Table 1: together with median show min-max, and together with mean value of BMI show standard
deviation.
Answer: We improved table 1 on page 6 as recommended.
Changed Manuscript:
Table 1 on page 6.
4. Table 1: you should statistically compare all subgroups in terms of all parameters.
Answer: We statistically assessed the relationship between CIN and age, gravida, BMI, menopause,
HPV, and conization size. And the statistical results were added on page 6 of the results section
and page 8 of the discussion section.
5. In all tables leave "p" instead of "p value".
Answer: We corrected “p” in Tables 2 and 3.
6. The limitations of the research should be presented at the end of Discussion.
Answer: We presented the limitations of our study in the discussion section on page 9.

Reviewer 3 Report
Introduction poorly presents the pathology of precancerous lesions of the cervix. At present we rather use the nomenclature of HSIL and LSIL lesions instead of the histological terms CIN, for purely clinical reasons - diagnostic schemes and treatment recommendations.
"reduce the number of patients who must undergo conical resection for diagnosis. 50" - it is not recommended to provide conisation for diagnostic purposes... We easily can provide biopsy for diagnostics...
"whether these differences in electrical properties can be exploited to noninvasively screen for and diagnose CIN. 72" - if at all, screening accuracy, not diagnosis...
"69 with suspected CIN who underwent loop electrical excision procedure (LEEP). 76" - what is the rationale to provide LEEP in CIN1/LSIL patients...???
- the analysis of the diagnostic potential of the method should be assessed in situ, not on the excised lesion; in regard to the tested applicability and disturbing factors appearing after the removal of the lesions; cold knife conization if at all ...
- from clinical point of view analysis non-CIN vs. all CINs (1+2+3) is useless... We are interested in differentiation between LSIL and HSIL... or HSIL vs. non-HSIL (treatment vs. no treatment...)
- confounding factors have not been analyzed in the statistics - HPV status, cervicitis, size, which may have strong impact on the results ...
"When normal tissue progresses to cervical dysplasia, the stromal contribution to tissue resistivity is influenced by factors such as extracellular hydration, matrix content, and cellular density [23]." - there is no ECM in epithelial tissue (except basement membrane), what is a matrix in epithelium? Reference is from 1998...
"...evaluation of cervical electrical properties can provide instant results. 288" - No. It may only improve screening proccess. For result we need histopathological examination.
Author Response
1. Introduction poorly presents the pathology of precancerous lesions of the cervix. At present we
rather use the nomenclature of HSIL and LSIL lesions instead of the histological terms CIN, for purely
clinical reasons - diagnostic schemes and treatment recommendations.
Answer: We entirely agree with your point. However, some clinicians are still more familiar with the CIN
classification system, and previous cervical cancer diagnosis studies using Electrical
Impedance Spectroscopy (EIS) have also used the CIN classification system. In our research,
the Bethesda classification system was used for histological classification because our method
is not a diagnostic tool but a screening tool. We developed the bioimpedance spectrum-based
multi-electrode probe as a primary screening tool to determine the stage of clinical management.
2. "reduce the number of patients who must undergo conical resection for diagnosis. 50" - it is not
recommended to provide conisation for diagnostic purposes... We easily can provide biopsy for
diagnostics...
Answer: Thank you for your comment. However, LEEP (Loop Electrosurgical Excision Procedure) is
also used as a diagnostic tool in addition to the therapeutic purpose. As a reference of NCCN
Guidelines Version 2.2012 Cervical Cancer Screening, it is used for diagnostic purposes when
the boundary of the lesion is not visible on colposcopy, when the squamous-columnar junction
(SCJ) is not visible on colposcopy, when an endocervical lesion is suspected, or when the
findings of Pap smear test, biopsy, and colposcopy are inconsistent. Therefore, it is also
possible to reduce the number of patients who must undergo conical resection for diagnosis in
these cases if the results of bioimpedance spectrum-based multi-electrode probe can clearly
present the clinical status.
3. "whether these differences in electrical properties can be exploited to noninvasively screen for and
diagnose CIN. 72" - if at all, screening accuracy, not diagnosis...
Answer: Thank you for your comment. Our purpose is to screen CIN using the bioimpedance spectrum-
based multi-electrode probe. We modified the sentence. Also, we clearly explained the purpose
of this study in the introduction.
Changed Manuscript:
On page 2, “Moreover, we evaluated whether these differences in electrical properties can be ex-ploited
to noninvasively screen for CIN.”
4. "69 with suspected CIN who underwent loop electrical excision procedure (LEEP). 76" - what is the
rationale to provide LEEP in CIN1/LSIL patients...???
Answer: Thank you for your comment. In this study, the group of CIN I was composed of patients who
were suspicious of CIN and finally diagnosed CIN I from LEEP. The group of patients diagnosed
with CIN I with LEEP was as follows.
1. Patients with a biopsy diagnosis of CIN I who refuse ECC (endocervical curettage) or
disagree with evaluation every six months.
2. Patients with ASC-H or HSIL in colposcopy but negative or CIN I on cervical biopsy
3. Patient with suspicious CIN II/III in colposcopy but their final diagnosis is CIN I by LEEP
4. Patients with suspicious lesions who have a phobia for cervical cancer and want a definitive
diagnosis
Reference: NCCN Guidelines Version 2.2012 Cervical Cancer Screening
5. the analysis of the diagnostic potential of the method should be assessed in situ, not on the excised
lesion; in regard to the tested applicability and disturbing factors appearing after the removal of the
lesions; cold knife conization if at all ...
Answer: We completely agree with your comment. This is the limitation of this study mentioned in the
discussion. However, It is the first preclinical study using the developed bioimpedance
spectrum-based multi-electrode probe. Before applying clinical study, this kind of research step
is required. We plan to apply it in-vivo clinical study as mentioned in the discussion based on
the results of ex-vivo obtained in this study.
6. from clinical point of view analysis non-CIN vs. all CINs (1+2+3) is useless... We are interested in
differentiation between LSIL and HSIL... or HSIL vs. non-HSIL (treatment vs. no treatment...)
Answer: Since all of the CIN II cases in our study were p16-negative, they were classified as LSIL.
However, the development of the bioimpedance spectrum-based multi-electrode probe was to
screen CIN as mentioned in answers 1 and 3. We described the future direction of research in
the discussion section as you recommended.
7. confounding factors have not been analyzed in the statistics - HPV status, cervicitis, size, which may
have strong impact on the results ...
Answer: We statistically assessed the relationship between CIN and age, gravida, BMI, menopause,
HPV, and conization size. And the statistical results were added on page 6 of the results section
and page 8 of the discussion section.
8. "When normal tissue progresses to cervical dysplasia, the stromal contribution to tissue resistivity is
influenced by factors such as extracellular hydration, matrix content, and cellular density [23]." - there
is no ECM in epithelial tissue (except basement membrane), what is a matrix in epithelium? Reference
is from 1998...
Answer: Thank you for your comment. We corrected the discussion based on the reference [9].
Changed Manuscript:
On page 8, “The development of cervical dysplasia from normal tissue is accompanied by structural
changes caused by an increased nuclear/cytoplasmic ratio, loss of the layer of flattened cells close to
the surface, and surface mucus and stromal tissue characteristics [9, 23]. The destruction of tissue
structure could reduce resistivity at low frequencies, making the difference between normal and CIN
samples particularly apparent at low frequencies [9]. The present study results confirmed that the
resistivity differences between non-CIN and CIN were more apparent in the frequency spectrum.”
9. "...evaluation of cervical electrical properties can provide instant results. 288" - No. It may only
improve screening process. For result we need histopathological examination.
Answer: We corrected this sentence.
Changed Manuscript:
On page 9, “Moreover, applying the bioimpedance spectrum-based multi-electrode probe can pro-vide
instant resistivity spectrum results for screening of CIN.”
Reviewer 4 Report
This manuscript reports tissue characterization using a multi-electrode probe based on electrical bioimpedance spectroscopy for screening cervical intraepithelial neoplasia. Determination of cervical electrical properties is a promising screening tool with similar sensitivity and specificity to currently used screening tests. Assessment of the electrical properties combined with other devices may provide additional and valuable information for real-time monitoring. The manuscript can be recommended for publication after minor revision of some parts of the section and some grammatical errors.
Author Response
This manuscript reports tissue characterization using a multi-electrode probe based on electrical
bioimpedance spectroscopy for screening cervical intraepithelial neoplasia. Determination of cervical
electrical properties is a promising screening tool with similar sensitivity and specificity to currently used
screening tests. Assessment of the electrical properties combined with other devices may provide
additional and valuable information for real-time monitoring. The manuscript can be recommended for
publication after minor revision of some parts of the section and some grammatical errors.
Answer: Thank you for your comment. We corrected the whole script of the paper, and a native speaker
checked grammatical errors.
Round 2
Reviewer 1 Report
The authors have considered all my previous comments in a satisfying way, and the quality of the manuscript has been consequently increased. I recommend publication.
Reviewer 3 Report
Ok. I see strong improvement of submitted paper. Moreover with explanations given by the authors some concerns were clarified.
Ad 1. All good but you cannot explain usage of CIN terminology instead of SIL that way. That's not our job to address scientific papers to those who are not up-to-date. I may understand that if you focused on scientific and technical issues in your paper you wanted to analyze in deep more different groups which can be somehow differentiated. That's your potential reason which might be mentioned...
Ad 2. If you provide such explanation reader will know why you performed non-recommended procedure, as it seems initially. Now we know. It must be clearly visible. Moreover, why you refer to the 2012 NCCN guidelines as we have more current?
Ad 3. OK
Ad 4. OK - good points which should be easily visible in the paper.
Ad 5. Ok - this answers my concerns.
Ad 6. It does not convince me... What is the clinical impact?... You may try to explain it as in the 1 point. Moreover, it should be than well presented that it is at the moment introduction to the in vivo studies and it might be treated as data collection study....
Ad 7. OK
Ad 8. OK
Ad 9. OK
Author Response
Please see the attachment response and revised paper.
